# RNA sequencing reveals niche gene expression effects of beta-hydroxybutyrate in primary myotubes

Philip MM Ruppert[1], Lei Deng[1], Guido JEJ Hooiveld[1], Roland WJ Hangelbroek[1,2], Anja Zeigerer[3,4], Sander Kersten[1]

**Various forms of fasting and ketogenic diet have shown promise in (pre-)clinical studies to normalize body weight, improve metabolic health, and protect against disease. Recent studies suggest that β-hydroxybutyrate (βOHB), a fasting-characteristic ketone body, potentially acts as a signaling molecule mediating its beneficial effects via histone deacetylase inhibition. Here, we have investigated whether βOHB, in comparison to the well-established histone deacetylase inhibitor butyrate, influences cellular differentiation and gene expression. In various cell lines and primary cell types, millimolar concentrations of βOHB did not alter differentiation in vitro, as determined by gene expression and histological assessment, whereas equimolar concentrations of butyrate consistently impaired differentiation. RNA sequencing revealed that unlike butyrate, βOHB minimally impacted gene expression in primary adipocytes, macrophages, and hepatocytes. However, in myocytes, βOHB up-regulated genes involved in the TCA cycle and oxidative phosphorylation, while down-regulating genes belonging to cytokine and chemokine signal transduction. Overall, our data do not support the notion that βOHB serves as a powerful signaling molecule regulating gene expression but suggest that βOHB may act as a niche signaling molecule in myocytes.**

## Introduction

Prevalence rates for obesity are spiraling out of control in many communities across the world. Inasmuch as obesity is a major risk factor for many chronic diseases, including type 2 diabetes, cardiovascular disease, and certain types of cancer (1), effective remedies to slow down the growth of obesity are direly needed. A common strategy that effectively promotes weight loss, at least in the short term, is caloric restriction, leading to an improvement in the cardiometabolic risk profile. One of the more popular forms of caloric restriction is time-restricted feeding, in which the normal abstinence of food consumption during the night is partly extended into the daytime (2). Other forms of caloric restriction include

alternate day fasting, periodic fasting (e.g., 5:2), and fasting mimicking diets (2). In animal models, these dietary interventions increase median life-span, reduce body weight, mitigate inflammation, improve glucose homeostasis and insulin sensitivity, and delay the onset of diabetes, cardiovascular and neurological disease, as well as cancer. Similarly, human studies have reported weight loss, reduced HbA1c and glucose levels, improved insulin sensitivity and blood lipid parameters, and lower blood pressure (2, 3, 4, 5, 6, 7).

Interestingly, it has been suggested that intermittent fasting may confer cardiometabolic health benefits independently of caloric restriction and concomitant weight loss (7, 8). A number of mechanisms have been invoked in explaining the possible health benefits of the various forms of fasting as well as of ketogenic diets, including lower plasma insulin levels and higher plasma levels of ketone bodies. Ketonemia is a characteristic feature of the fasted metabolic state. During the feeding–fasting transition, the body switches from glucose as a primary fuel source to the oxidation of fatty acids. In the liver, the high rates of fatty acid oxidation are accompanied by the synthesis of ketone bodies, which, as fasting progresses, become the dominant fuel for the brain (9). The two main ketone bodies are β-hydroxybutyrate (βOHB) and acetoacetate (AcAc). Both compounds serve as sensitive biomarkers for the fasted state, increasing in combined concentration from less than 0.1 mM in the fed state to 1 mM after 24 h to 5–7 mM when fasting for about a week (9, 10, 11).

In addition to serving as fuel in tissues such as the brain, heart, and skeletal muscle, recent research has unveiled that βOHB may also serve as a direct signaling molecule. By activating specific signaling pathways, βOHB may not only have an important regulatory role in the metabolic response to fasting but may also potentially mediate some of the beneficial health effects of fasting (2, 12, 13, 14, 15, 16, 17, 18, 19, 20, 21, 22, 23). Evidence has been presented that βOHB may regulate gene expression via epigenetic mechanisms. Shimazu et al linked βOHB-mediated HDAC inhibition to protection against oxidative stress in the kidneys via the up-regulation of *FOXO3a*, *Catalase*, and *MnSOD* (24). Whereas subsequent studies in neonatal hepatocytes, brain microvascular endothelial cells, and NB2a neuronal cells hinted at conservation of

[1]Nutrition, Metabolism and Genomics Group, Division of Human Nutrition and Health, Wageningen University, Wageningen, The Netherlands [2]Euretos BV, Utrecht, The Netherlands [3]Institute for Diabetes and Cancer, Helmholtz Center Munich, Neuherberg, Germany and Joint Heidelberg-Institute for Diabetes and Cancer Translational Diabetes Program, Inner Medicine 1, Heidelberg University Hospital, Heidelberg, Germany [4]German Center for Diabetes Research (DZD), Neuherberg, Germany

Correspondence: sander.kersten@wur.nl

this pathway in different cell types (25, 26), other studies have since questioned the role of βOHB as a potential physiological HDAC inhibitor (27, 28). Interestingly, recent studies in hepatocytes, cortical neurons, myotubes, and endothelial cells suggested that βOHB may serve as a novel substrate for transcriptionally activating histone modifications. This so-called lysine β-hydroxybutyrylation was found in proximity to fasting-relevant hepatic pathways, including amino acid catabolism, circadian rhythm, and PPAR signaling (28), and was found to regulate the expression of BDNF (29) and hexokinase 2 (27). How histones become β-hydroxybutyrylated remains unknown but a series of biochemical experiments suggest that SIRT3 facilitates the de-β-hydroxybutyrylation of histones (30). While there is thus some evidence to suggest that βOHB may serve as a direct signaling molecule regulating genes, the potency and importance of βOHB as regulator of gene expression in various cell types is unclear. Accordingly, here we aimed to investigate the capacity of βOHB to regulate gene expression and thereby serve as a direct signaling molecule during the fasted state. To this end, we investigated whether βOHB, in comparison to the well-established HDAC inhibitor butyrate, influences in vitro differentiation of adipocytes, macrophages, and myotubes. In addition, we studied the effect of βOHB on whole genome gene expression in primary mouse adipocytes, macrophages, myotubes and hepatocytes via RNA-seq.

# Results

## Butyrate but not β-hydroxybutyrate impairs differentiation of adipocytes, monocytes, and macrophages

To solidify the concept of βOHB being a powerful signaling molecule that influences cellular homeostasis, we examined whether βOHB affects cellular differentiation. Previously, we showed that butyrate, despite acting as a selective PPARγ agonist, inhibits adipogenesis in 3T3-L1 cells (31). Because of structural and possibly functional resemblance with butyrate, we hypothesized that βOHB might exert similar effects on the differentiation of 3T3-L1 cells. Compared with the control, 8 mM βOHB did not visibly affect adipocyte differentiation, as assessed during the differentiation process (Day 4) and terminally (Day 10; Fig 1A). By contrast and in line with previous studies, 8 mM butyrate markedly inhibited adipocyte differentiation (Day 4 and 10; Fig 1A), whereas 1 μM rosiglitazone stimulated the differentiation process (Day 4). Corroborating the visual assessment, butyrate significantly down-regulated the expression of the adipogenic marker genes Adipoq, Slc2a4 (Glut4), and Fabp4, whereas rosiglitazone significantly up-regulated these genes. In line with the lack of effect of βOHB on 3T3-L1 differentiation, βOHB had a minor impact on the expression of Slc2a4 (Glut4) and no impact on the expression of Adipoq or Fabp4 (Fig 1B).

Next, we studied myogenesis. Butyrate was previously reported to inhibit myogenesis when present during the differentiation process (32). To assess whether βOHB might influence myogenesis, we differentiated C2C12 myoblasts in the presence of 5 mM βOHB or 5 mM butyrate. In line with previous reports, butyrate inhibited the differentiation of myoblasts towards myotubes (Fig 1C) (32). By

contrast, βOHB did not visibly impact myotube formation (Fig 1C). Myogenesis is driven by muscle regulatory factors including MyoG, MyoD, and Myf5 (33, 34). Supporting the lack of effect of βOHB on myogenesis, expression levels of all three muscle regulatory factors were similar in βOHB and control-treated C2C12 cells at any time-point during the differentiation process (Fig 1D). This is in clear contrast to the treatment with butyrate, which prevented up-regulation of MyoG and MyoD and down-regulated Myf5 at all time points, respectively. We also wondered whether instead of influencing the differentiation process, βOHB might affect the polarization of myotubes to either myosin heavy chain class I (MHCI) or class II (MHCII). Expression of Myh3, Myh7, and Myh8, representing MHCI, was unchanged between βOHB and control-treated myoblasts. Expression of Myh1, Myh2, and Myh4, representing MHCII, was also unchanged between βOHB and control (Fig S1), suggesting that βOHB does not influence the polarization of myotubes.

Furthermore, βOHB and butyrate have been reported to modulate immune cell function and viability (19, 35). Specifically, butyrate demonstrated pro-apoptotic effects on THP-1 in previous studies (36, 37, 38). To assess whether either compound influences the differentiation of a monocytic cell line in vitro, we differentiated THP-1 cells with PMA in the presence of 8 mM βOHB or butyrate. Corroborating reports of pro-apoptotic effects of butyrate on THP-1 cells (36, 37, 38), addition of butyrate during the differentiation process resulted in a clear reduction in the density of monocytes (Fig 1E). In keeping with the lack of effect on myocyte and adipocyte differentiation, βOHB also did not visually impact THP-1 cell differentiation (Fig 1E). PMA-induced differentiation of THP-1 cells is marked by differential expression of several marker genes including CD11b, CD14, TNF-α, and CD68 (39, 40, 41, 42). Butyrate prevented PMA-mediated induction of CD11b and CD68, and further increased TNF-α, CD14, and IL-1β expression (Fig 1F). In addition, butyrate markedly suppressed the expression of the pattern recognition receptor TLR4a and TLR4b and the lipid-associated genes LPL and CD36. By contrast, gene expression changes by βOHB for most genes were non-significant relative to cells treated with PMA only (Fig 1F). Interestingly, βOHB significantly altered gene expression of CD11b, CD14, LPL, and IL-1β, although the magnitude of the effect was modest (Fig 1F). These results suggest that butyrate exerts a strong effect on the differentiation and viability of THP-1 cells. In comparison, the effects of βOHB are small. Last, we wondered whether the effects observed for βOHB and butyrate would also translate to primary cell types. In primary mouse adipocytes differentiated from the stromal vascular fraction (SVF) and human monocytes obtained from buffy coats, we found similar effects as described for the immortalized cell lines. βOHB had negligible effects on the differentiation of primary adipocytes and primary human monocytes and on qRT-PCR readouts of differentiation marker genes (Fig 1G–I). Conversely, butyrate significantly inhibited the differentiation of primary mouse adipocytes and repressed the expression of differentiation marker genes (Fig 1G–I). Higher concentrations of butyrate but not βOHB also reduced cell density of primary human monocytes over time (data not shown), resembling the pro-apoptotic effects in THP-1 monocytes. Together, these results suggest that the differential effects of βOHB and butyrate are conserved in an array of cell types from immortalized and primary sources.

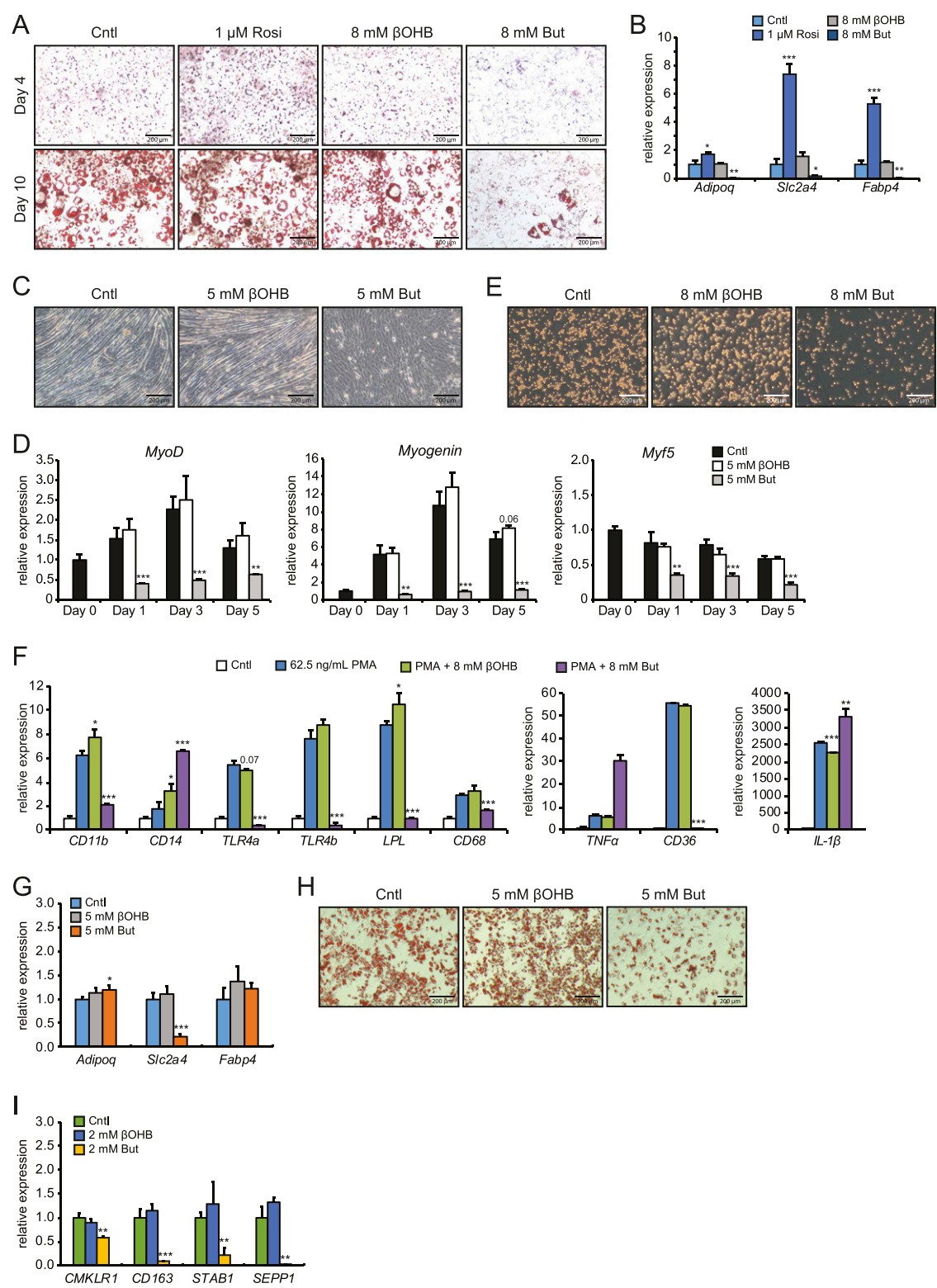

**Figure 1.   Differential effects of βOHB and butyrate on the differentiation process of adipocytes, myotubes, and macrophages.**
**(A)** Representative Oil red O staining of 3T3-L1 adipocytes at Day 4 of the standard differentiation protocol in the presence of either 1 µM Rosi, 8 mM βOHB, or 8 mM butyrate. **(B)** Corresponding expression profile of differentiation markers and PPARγ targets determined by RT-qPCR at Day 4 using the mild differentiation protocol. **(C)** Representative microscopic pictures of C2C12 myotube formation after 5 d of differentiation in the presence of 5 mM βOHB or 5 mM butyrate. **(D)** Corresponding expression profile of myocyte differentiation markers *MyoD*, *Myogenin*, and *Myf5* after differentiation. **(E)** Representative pictures of THP-1 cells differentiated for 24 h in 62.5 ng/ml PMA in presence of either 8 mM βOHB or 8 mM butyrate. **(F)** Corresponding expression profile of THP-1 differentiation markers. **(G)** Representative Oil red O staining of primary

### β-hydroxybutyrate alters gene expression in primary myocytes but not primary adipocytes, macrophages, and hepatocytes

We reasoned that if βOHB has a signaling function, it would likely alter the expression of genes either directly or indirectly. Accordingly, we investigated the ability of βOHB to regulate gene expression in cells that have been suggested to be targeted by βOHB. Specifically, we collected primary mouse adipocytes, primary mouse BMDMs, primary mouse myotubes, and primary mouse hepatocytes and performed RNA sequencing after 6 h treatment with either 5 mM βOHB or 5 mM butyrate. Importantly, the RNAseq data confirmed that all cell types expressed at least one type of the monocarboxylate transporters *Slc16a1* (*Mct1*), *Slc16a7* (*Mct2*), and *Slc16a6* (*Mct7*), which are responsible for the transport of βOHB and butyrate (19, 43, 44, 45) (Figs 2A and S2A). In line with uptake and utilization, we observed a 50% reduction of βOHB medium levels in primary adipocytes over a 3-d culture period (Fig S2B).

The cells treated with butyrate showed an anti-conservative *P*-value distribution, suggesting that butyrate has a marked effect on gene expression in all cell types studied. Conversely, cells treated with βOHB showed a uniform or conservative *P*-value distribution (Fig S2C), suggesting that βOHB treatment minimally impacted gene expression. To study the magnitude of gene regulation by βOHB and butyrate in the various primary cells, we performed volcano plot analysis. Strikingly, the effect of βOHB on gene expression was very limited in all cell types, with only a small number of genes reaching the statistical threshold of $P < 0.001$ (Fig 2B). Using this statistical threshold, βOHB significantly altered expression of 44, 38, 466 and 95 genes in adipocytes, macrophages, myocytes and hepatocytes, respectively. Of these genes, 20, 13, 388, and 32 were down-regulated, respectively (Fig 2C). In adipocytes, macrophages, and hepatocytes, less than 10 genes had a false discovery q-value below 0.05, indicating that most of the significant genes in these cells likely represent false positives. In myocytes, 560 genes had a FDR q-value below 0.05 (Fig S2D). In stark contrast to the relatively minor effects of βOHB, butyrate had a huge effect on gene expression in all primary cells (Fig 2B). Butyrate significantly changed the expression of 7,068, 7,943, 6,996 and 7,158 genes in adipocytes, macrophages, myocytes and hepatocytes, respectively ($P < 0.001$), of which 50–52% were down-regulated (Fig 2C). The number of differentially expressed genes was similarly high when using a FDR q-value of 0.05 (Fig S2D).

To further examine the overall effect of βOHB and butyrate on gene regulation in the various cell types, we performed hierarchical clustering and principle component analysis. Both analyses showed that the samples cluster by cell type first. Whereas the butyrate-treated samples clustered apart from the control and βOHB samples in each cell type, the control and βOHB samples did not cluster separately from each other (Figs 2D and E and S2E). Collectively, these data indicate that in comparison to butyrate, βOHB minimally impacted gene expression in adipocytes, macrophages, and hepatocytes. By contrast, βOHB had a more pronounced effect on gene expression in myocytes, although still much less than observed for butyrate.

### Significant overlap in gene regulation by butyrate across various cell types

Next, we studied the similarity in gene regulation by butyrate among the different cell types. Hierarchical biclustering of all significantly regulated genes per condition showed marked similarity in the response to butyrate. Furthermore, Venn diagrams for the butyrate-treated cells revealed that a large fraction of the significantly regulated genes were shared in all cell types, confirming the similarity in gene regulation by butyrate. Indeed, 18% (1,250 genes) of all significantly up-regulated genes were up-regulated in every cell type. Similarly, 15% (1,095 genes) of all significantly down-regulated genes were down-regulated in every cell type (Fig 3A). Heat maps of the top 20 most significantly regulated genes by butyrate showed comparable signal log ratios in all four cell types (Fig 3B). qRT-PCR analysis for a few selected genes confirmed regulation by butyrate (Fig S2F). Interestingly, the heat maps for butyrate lists several genes related to histone metabolism (*H1f0*, *H1f2*, *H1f4*, *H1f3*, *Hcfc1*, *Phf2*, and *Anp32b*).

To examine the similarity in gene regulation by butyrate across the different cell types at the level of pathways, we performed gene set enrichment analysis (GSEA) using the top 100 up- and down-regulated genes according to the T-statistic. The overlap in significantly regulated pathways (FDR q < 0.1) are shown in a Venn diagram, revealing a high overlap for butyrate-induced and repressed pathways among the four cell types. 20 out of 61 pathways were induced in at least two cell types, while the four pathways ("phosphatidylinositol-signaling-system" "inositol-phosphate-metabolism" "arginine-and-proline-metabolism," and "fatty-acid-elongation") were induced in all four cell types (Fig 3C). Interestingly, 33 pathways were exclusively induced by butyrate in myocytes (Fig 3C). Conversely, 19 of 52 pathways were repressed in at least two cell types, whereas the three pathways ("spliceosome," "chronic-myeloid-leukemia" and "bladder-cancer") were repressed in all four cell types (Fig 3C). Plotting the top 10 induced and repressed pathways by average normalized enrichment scores corroborates the consistent regulation of pathways by butyrate among the various cell types (Fig 3D). Collectively, these analyses indicate considerable overlap in the effect of butyrate on gene expression in all cell types at the gene and pathway level.

### Significant effect of βOHB on gene regulation in primary myocytes

Given the minimal number of genes altered by βOHB in adipocytes, macrophages, and hepatocytes, most likely representing false positives, we did not further perform any analyses for these cell types. Instead, we focused our attention on the effects of βOHB on gene regulation in myocytes. Having noted a region of overlap between βOHB and butyrate (Fig 2E; black rectangles), we first investigated the similarity in gene regulation between both compounds in myocytes. Venn diagram analysis revealed that of the 451 genes down-regulated by βOHB according to FDR q < 0.05, 320 genes

adipocytes at Day 7 of the standard differentiation protocol with either 5 mM βOHB or 5 mM butyrate. **(H)** Corresponding expression profile of adipogenesis differentiation markers as determined by RT-qPCR at Day 4 of differentiation. **(I)** Gene expression of differentiation markers for human primary monocytes after 7-d culture in M-CSF with either 2 mM βOHB or 2 mM butyrate. Error bars represent SD. Asterisks indicate significant differences according to *t* test compared with control (of respective day) or PMA treatment (*$P < 0.05$; **$P < 0.01$; ***$P < 0.001$).

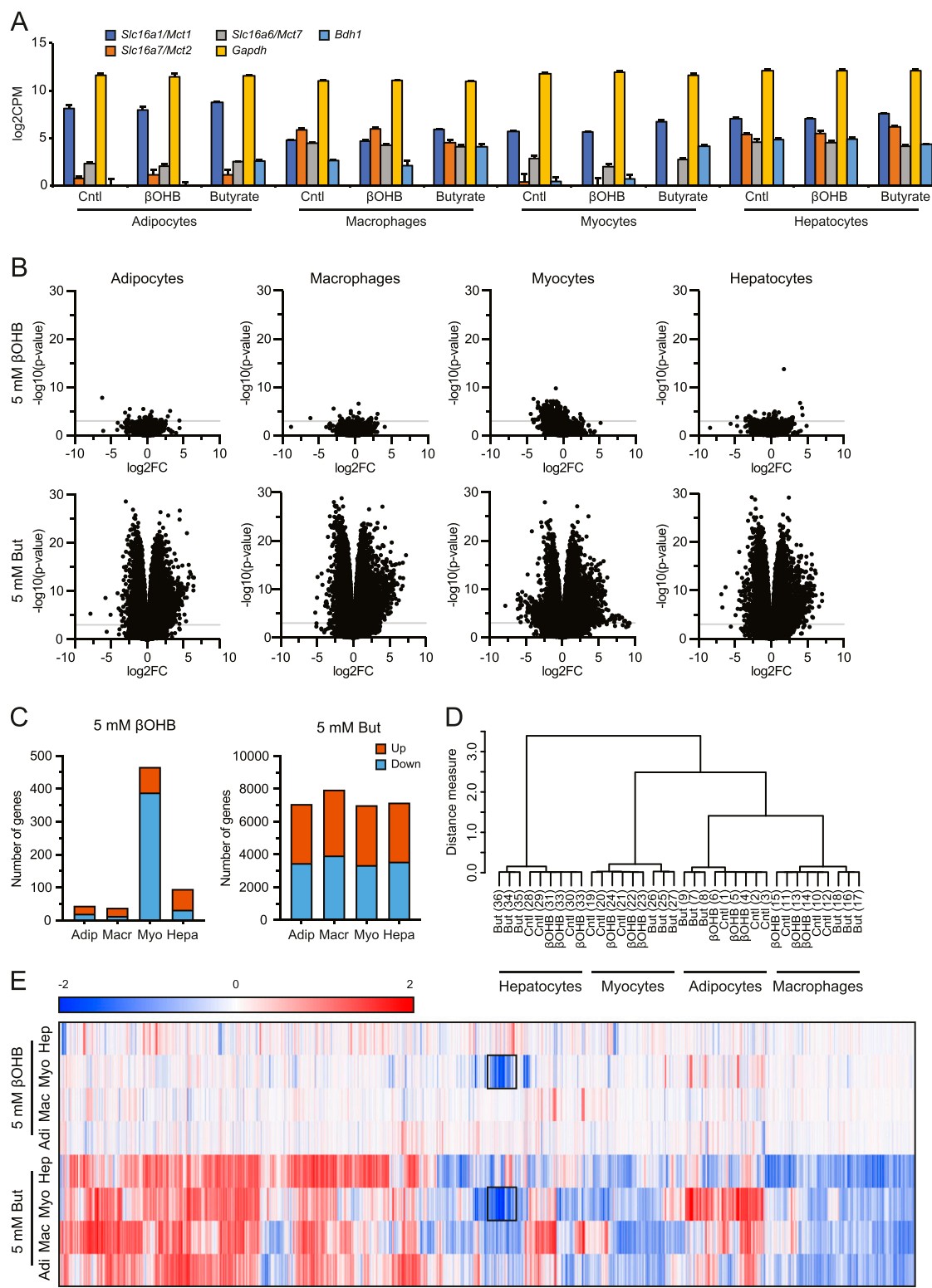

**Figure 2. Disparate effects of βOHB and butyrate on gene expression in primary adipocytes, macrophages, myocytes, and hepatocytes.**
**(A)** Expression levels (log$_2$CPM) of *Bdh1* and monocarboxylate transporters *Mct1*, *Mct2*, and *Mct7* in relation to *Gapdh*. **(B)** Volcano plots showing log$_2$[fold-change] (x-axis) and the −10log of the raw *P*-value (y-axis) for every cell type treated with βOHB and butyrate. The grey line indicates *P* = 0.001. **(C)** Number of genes significantly (*P* < 0.001) altered by treatment with βOHB and butyrate. **(D)** Hierarchical clustering of βOHB and butyrate-treated samples. **(E)** Hierarchical biclustering of βOHB and butyrate-treated samples visualized in a heat map. Clustered are significant differentially expressed genes based on Pearson correlation with average linkage. Red indicates up-regulated, blue indicates down-regulated. Black rectangle marks genes that appear similarly regulated by βOHB and butyrate in myocytes.

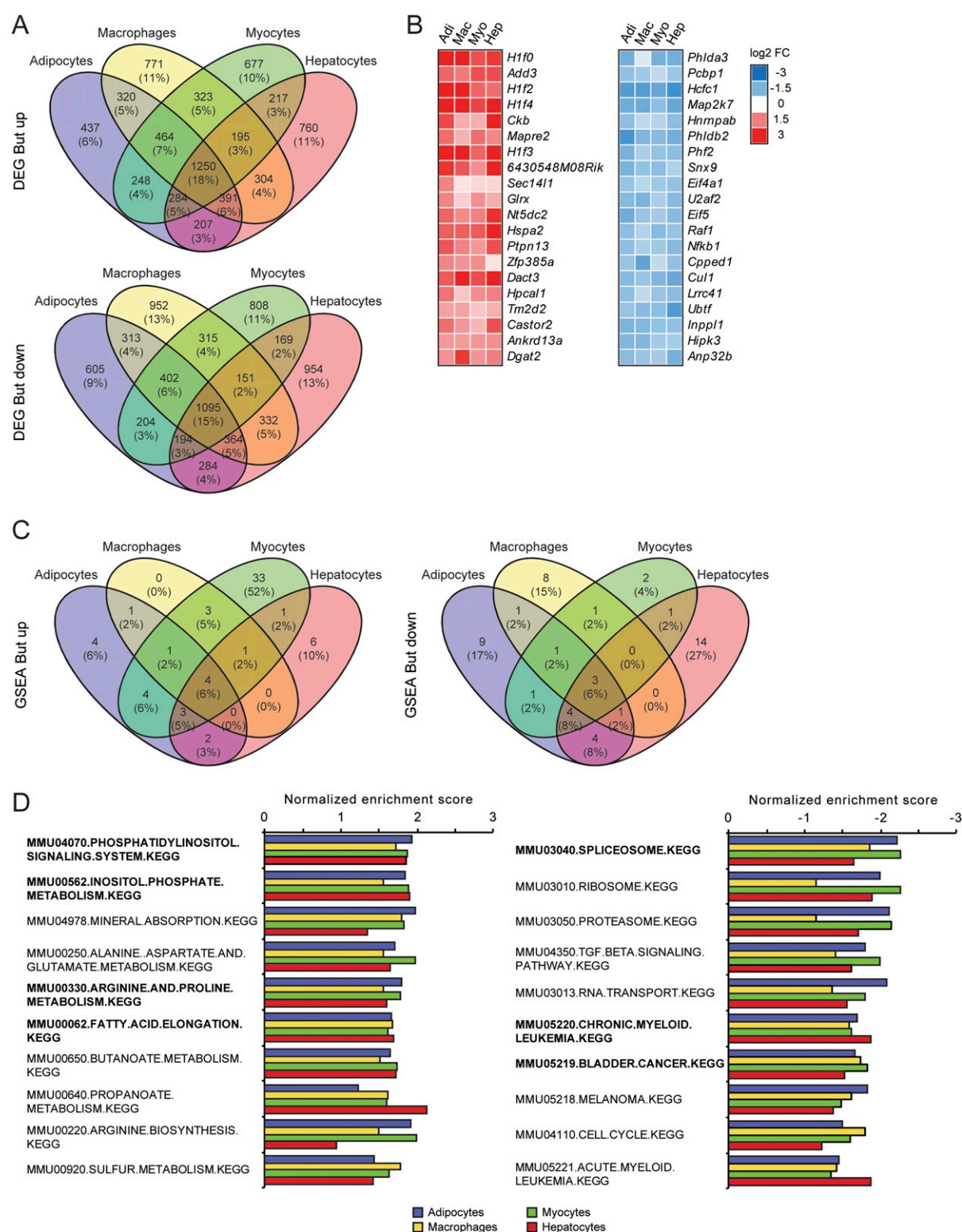

**Figure 3.   Consistency of gene expression changes elicited by butyrate.**
**(A)** Venn diagrams showing overlap in significantly regulated genes by butyrate between cell types ($P < 0.001$), separated into up- and down-regulated genes. **(B)** Heat maps showing genes that are significantly up- or down-regulated by butyrate in all four cell types ($P < 0.001$). Genes are sorted by statistical significance. **(C, D)** Top 10 up- and down-regulated gene sets in βOHB (C) and butyrate-treated cells (D). Gene sets were determined by gene set enrichment analysis based on t-values and are ranked according to averaged normalized enrichment score. Pathways in bold are significantly enriched in all four cell types.

(71%) were also significantly down-regulated by butyrate. Likewise, 50% of the 109 genes up-regulated by βOHB were also up-regulated by butyrate (Fig 4A). Table S1 shows a list of genes regulated by βOHB according to FDR $P < 0.001$. qRT-PCR analysis for *Clec4a1* and *Apoc2* confirmed their down-regulation by βOHB (Fig S2G).

To further examine the similarity in gene regulation between βOHB and butyrate, we plotted $log_2Fc$ values of all genes in a correlation plot. The correlation plot showed considerable overlap in gene regulation between βOHB and butyrate, which was most obvious for the genes down-regulated by the two treatments (Fig 4B). To statistically analyze the overlapping gene regulation, we performed overlap analysis (46, 47). In this analysis, the expected overlap is calculated for any number of top genes (on the x-axis) using a hypergeometric distribution (i.e., overrepresentation analysis). The blue line and shaded blue area cover the expected overlap under the null hypothesis (95% CI), while the black line indicates the observed overlap (Fig 4C). Consistent with the Venn diagram and scatter plot, significant overlap was observed between βOHB and butyrate for the down-regulated genes but not for the up-regulated genes. Collectively, this may indicate a similar mode of action for both compounds.

To gain further insight into the pathways regulated by βOHB in myocytes, we performed GSEA and Enrichr analysis, first focusing on the up-regulated pathways. Using a statistical threshold of q < 0.1, GSEA yielded 25 gene set that were significantly up-regulated by βOHB in myocytes (Fig 4D and Table S2). Many of the up-regulated gene sets were related to metabolic pathways, including the TCA cycle, oxidative phosphorylation, and amino acid metabolism. Enrichr analysis ("WikiPathways Mouse") on the 78 up-regulated genes that met the statistical significance threshold of $P < 0.001$ yielded only one significant pathway (adjusted $P < 0.05$), which was TCA cycle (not shown). The top 40 list of most highly up-regulated genes presents a diverse set of genes involved in cell cycle progression, tissue and cell remodeling as well as gene regulation (Fig 4E).

With respect to down-regulation of gene expression, using a statistical threshold of q < 0.1 for the GSEA analysis, 96 gene sets were significantly down-regulated by βOHB in myocytes (Table S3). Many of the down-regulated pathways were related to immunity and inflammation (Fig 4D). Enrichr analysis ("WikiPathways Mouse") confirmed the enrichment of inflammation-related pathways (Fig 5A). The down-regulation of genes involved in immunity and inflammation was reflected in the top 40 list of most highly down-regulated genes (Fig 4E). The majority of these genes were similarly down-regulated by βOHB and butyrate, suggesting a common mechanism of regulation.

Last, to substantiate the notion that βOHB and butyrate might affect gene expression via a common mechanism, we plotted $log_2Fc$ values for all genes significantly down-regulated by βOHB in a correlation plot and determined the number of genes that fell within a fold-change ratio of 0.75× to 1.25×. Approximately 40–50% of all βOHB DEGs and of genes commonly regulated by βOHB and butyrate fell within this artificial cutoff, indicating that a substantial number of genes regulated by βOHB were regulated by butyrate to a similar extent (Fig 5B). Enrichr analysis of βOHB-downregulated genes for "Encode Histone modifications" and "DSigDB" showed significant overlap with gene signatures belonging to histone

modification experiments and treatments with common HDAC inhibitors, including vorinostat, valproic acid, and trichostatin A (Fig 5C). These data suggest that, in accordance with butyrate's well-established HDAC inhibitory function (48), βOHB may also regulate target genes via epigenetic mechanisms in primary myocytes.

# Discussion

In this work, we studied the potential of β-hydroxybutyrate (βOHB) to influence cellular differentiation and for the first time performed whole genome expression analysis in primary adipocytes, macrophages, myocytes, and hepatocytes comparing βOHB side-by-side with the well-established HDAC inhibitor butyrate. At physiologically relevant plasma concentrations of βOHB as measured after fasting or ketogenic diet, βOHB did not affect the differentiation of 3T3-L1 adipocytes, C2C12 myotubes, THP-1 macrophages, primary mouse adipocytes, and primary human monocytes. Furthermore, acute βOHB treatment minimally influenced gene expression in primary adipocytes, macrophages, and hepatocytes but altered the expression of a substantial number of genes in primary myocytes. The results from βOHB are in stark contrast to the consistent inhibition of differentiation by butyrate in 3T3-L1, C2C12, THP-1, primary mouse adipocytes, and primary human monocytes, and the profound and consistent gene expression changes caused by butyrate in the various primary cells. Together, these data do not support the notion that βOHB serves as a potent signaling molecule regulating gene expression in adipocytes, macrophages, and hepatocytes. The suppressive effect of βOHB in myocytes on the expression of genes involved in immunity merits further study.

Interest in ketones has surged in the recent years. Illustrated by the sheer abundance of reviews and perspective articles on the potential benefits of ketosis, βOHB is considered as a potential mediator of the putative fasting-related health benefits (2, 12, 13, 14, 15, 16, 17, 18, 19, 20, 21, 22, 23). Common to all reviews is the prominent portrayal of βOHB as a potent HDAC inhibitor influencing gene expression, a notion originating from work by Shimazu et al in kidney and HEK293 cells (24). In that study, evidence was presented that βOHB is an endogenous and specific inhibitor of class I histone deacetylases in vitro and in vivo, leading to protection against oxidative stress. However, recent studies have since been unable to confirm a HDAC inhibitory activity for βOHB in various cell types, using butyrate as positive control (27, 28, 29). Irrespective of the precise mechanism, epigenetic alterations ultimately require changes in gene expression to impact homeostasis. In our differentiation experiments, co-incubation with βOHB did not alter expression of key differentiation genes in 3T3-L1, C2C12, and THP-1 cells. The studies in primary mouse adipocytes and human primary monocytes corroborate these findings. Further unbiased assessment of whole genome expression in mouse primary cells revealed minimal effects of βOHB on gene expression in adipocytes, macrophages, and hepatocytes. In fact, we suspect that all genes significantly altered by βOHB in these cells represent false positives. Assuming that βOHB is taken up by hepatocytes, adipocytes, and macrophages, these results contradict the notion that βOHB acts as a general HDAC inhibitor.

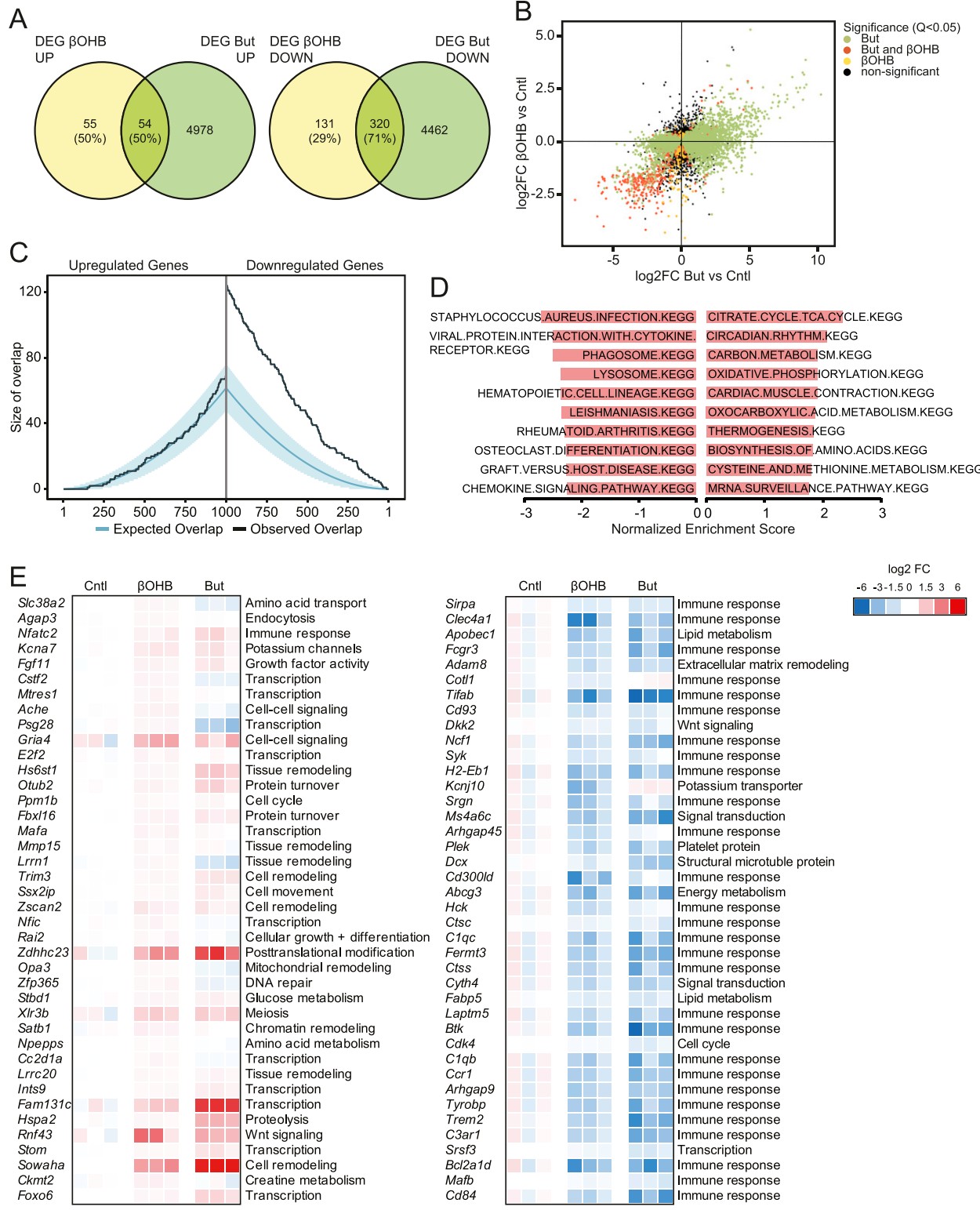

**Figure 4.  βOHB regulates genes and pathways related to TCA cycle and immunity in primary myocytes.**
**(A)** Venn diagrams showing overlap of significantly regulated genes by βOHB and butyrate, separated into up- and down-regulated genes ($P < 0.001$). **(B)** Correlation plot of gene regulation by βOHB and butyrate in myocytes. **(C)** Overlap plot depicting the size of the overlap for genes up-regulated (left) or down-regulated (right) by βOHB and butyrate treatment. The size of the overlap for randomly selected gene sets is shown by the blue line (blue shading depicts confidence interval). The observed overlap is shown by the black line. **(D)** Gene sets negatively enriched for βOHB treatment in myocytes according to gene set enrichment analysis. Gene sets are ranked according to Normalized Enrichment Score. **(E)** Heat maps showing top 40 up- and down-regulated genes by βOHB in primary myotubes, alongside butyrate.

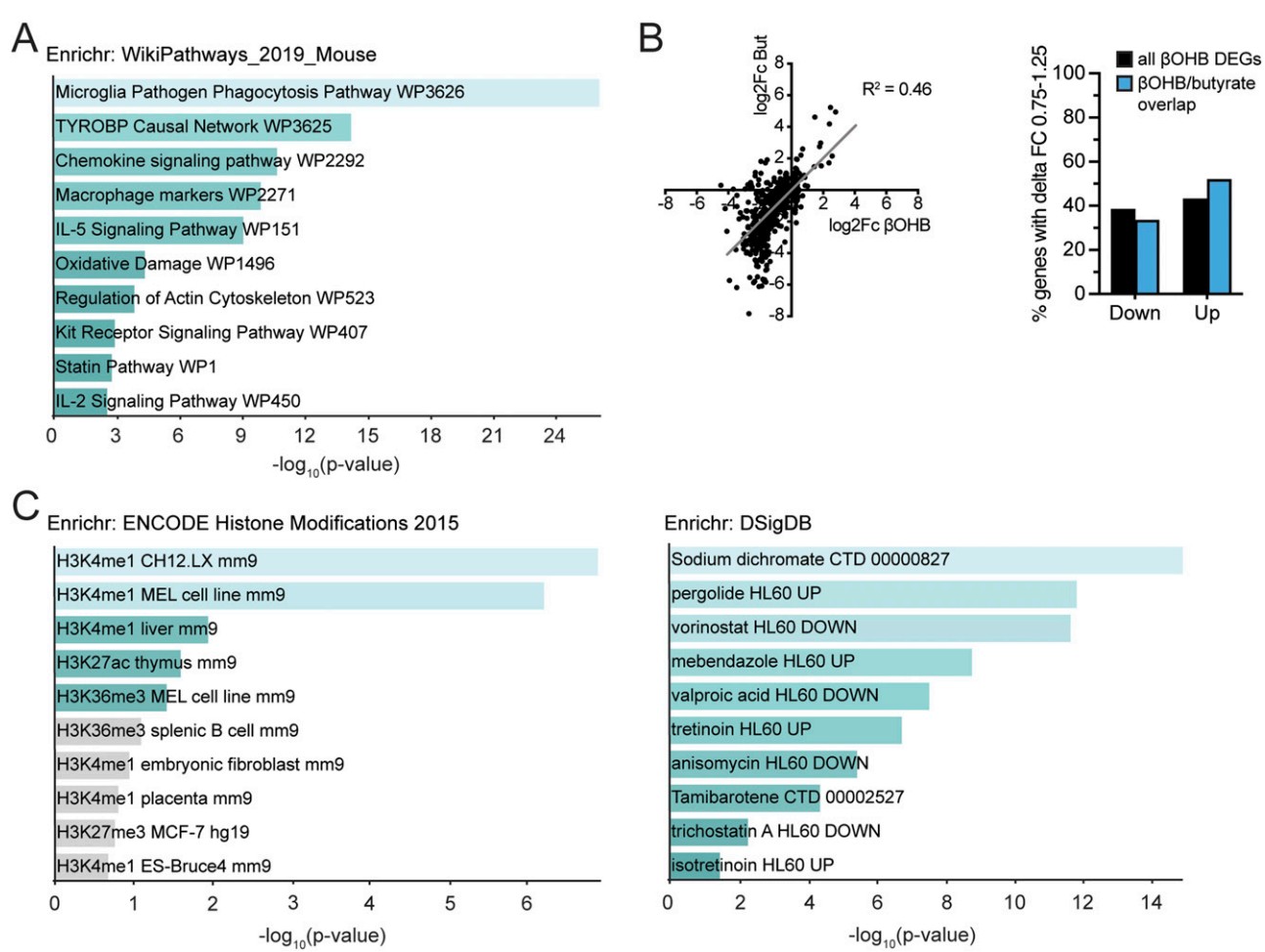

**Figure 5. βOHB-DEGs in myocytes are potentially regulated via epigenetic mechanisms.**
**(A)** Enrichr analysis of βOHB down-regulated genes (*P* < 0.001) according to "WikiPathways 2019 mouse." **(B)** Quantitation of genes that βOHB or βOHB and butyrate (overlap) regulated more similarly (FC ratio between 0.75× and 1.25×) split into up and down-regulated genes. **(C)** Enrichr analysis of βOHB down-regulated genes (*P* < 0.001) according to "ENCODE Histone modifications" and "DSigDB." Significance in Enrichr analyses: turquoise colored bars indicate *P* < 0.05.

An interesting finding of this study was that βOHB had distinct effects on gene expression in primary myotubes. Supporting the use of βOHB in muscle tissue as a substrate for ATP synthesis (49, 50), pathways related to TCA cycle and oxidative phosphorylation were up-regulated by βOHB. In addition, βOHB markedly influenced immunity-related pathways and specifically down-regulated various genes belonging to cytokine and chemokine signal transduction, including *Sirpa*, *Clec4a1*, *Fcgr3*, *Cd93*, *Syk*, *Ms4a6c*, *Hck*, *C1qc*, *Btk*, *C1qb*, and *Ccr1*. Considering that the *Mct* transporter expression profile is similar among the primary cells, it is unclear why βOHB only exerted these effects in myocytes and not, for example, in macrophages. Nevertheless, one could speculate that the down-regulation of immune-related pathways in muscle cells by βOHB may be part of a broader mechanism to suppress immunity during starvation. Indeed, it is well recognized that starvation presents a trade-off between, on the one hand, saving energy to prolong survival and, on the other hand, investing a sufficient amount of energy to maintain immune defenses. It can be hypothesized that βOHB may serve as a signaling molecule that mediates the suppressive effect of starvation on specific immune-related processes

(51, 52). Interestingly, although not supported by the results in adipocytes, macrophages, and hepatocytes, Enrichr analysis does hint at an epigenetic mode of action for βOHB in myocytes. Further studies will need to expand on the tissue-specific effects of βOHB and probe the functional significance of above-mentioned findings with in vivo knockout studies.

In contrast to βOHB, the effects of butyrate on gene expression were prominent and displayed consistency between the tested primary cell lines and the differentiation experiments. A significant portion of histone metabolism-related genes were consistently regulated between the various cell types. In addition, the most highly enriched pathways were significantly enriched in most if not all cells. In line with butyrate's well-established effects on gene expression, pathways relevant to transcriptional activities were significantly enriched. Additional analyses using Enrichr are in support of butyrate's prominent HDAC inhibitory action. The marked effect of butyrate on adipocyte and myocyte differentiation in 3T3-L1 and C2C12 cells is in line with previous research (31, 32) and may also partly be explained by epigenetic mechanisms (53). It should be noted, however, that the data presented here are not suitable to deduce potential

physiological effects of butyrate in vivo. Juxtaposing the supra-physiological concentration of 5 mM used in this study are reports of 1–12 and 14–64 $\mu$M butyrate in the peripheral and central blood circulation measured in sudden death victims (54).

The main limitation of our study is the exclusive utilization of in vitro systems. We opted for this approach to allow for the identification of target genes that may be consistently regulated in more than one cell type in a controlled environment. Although novel target genes would have to be replicated in vivo, this approach seemed more reasonable for this purpose than in vivo systems in which it is impossible to study the transcriptional regulation specifically attributable to $\beta$OHB. For example, the hepatic response to fasting is shaped by the fatty acid-PPAR$\alpha$ axis, which regulates nearly every branch in lipid metabolism and is indispensable for the physiological adaptation to fasting (55, 56). The increase of ketone body levels during fasting occurs concurrently with many other metabolic and hormonal changes, including increased plasma fatty acids, cortisol, and glucagon levels, and decreased plasma insulin and leptin levels.

In conclusion, this work for the first time systematically assesses the potential of the ketone body $\beta$OHB to influence gene expression in various primary cell types by RNA sequencing. The lack of genes commonly regulated among the various cell types coupled to generally insignificant effects on gene expression—with the exception of myocytes—contradict the notion that $\beta$OHB serves as a powerful and general signaling molecule regulating gene expression during the fasted state in vivo. Instead our data support the idea that $\beta$OHB acts as a niche signaling molecule regulating specific pathways in specific tissues such as muscle. Mechanistically, this action may include gene expression changes potentially via epigenetic effects but could also be secondary to oxidation or receptor activation. Collectively, in our view, the data presented here do not support the current portrayal of $\beta$OHB in the literature as the do-it-all-substrate during the fasted state and suggest that $\beta$OHB's effects may be much more nuanced and context-specific. Future research is necessary to delineate the role of $\beta$OHB including the regulation of gene expression in a tissue/context-specific manner, as for example, in muscle tissue.

## Materials and Methods

### Materials

$\beta$OHB was (R)-(−)-3-hydroxybutyric acid sodium salt from Sigma-Aldrich (#298360). Butyrate was Sodium butyrate from Sigma-Aldrich (#303410).

### Differentiation experiments

3T3-L1 fibroblasts were maintained in DMEM supplemented with 10% newborn calf serum and 1% penicillin/streptomycin (P/S) (all Lonza). Experiments were performed in six-well plates. For Oil red O staining, cells were differentiated using the standard protocol. 2 d post-confluence, cells were switched to DMEM supplemented with 10% FBS, 1% P/S, 0.5 mM isobutylmethylxanthine, 1 $\mu$M dexamethasone,

and 5 $\mu$g/ml insulin for 2 d in the presence of either 8 mM $\beta$OHB or 8 mM Butyrate. After 2 d, the cells were switched to DMEM supplemented with 10% FBS, 1% P/S, 5 $\mu$g/ml insulin, and the tested compounds for another 2 d. Then cells were maintained in normal DMEM medium (2–3 d), in the presence of the tested compounds until Oil red O staining on Day 10. Oil red O staining was performed following standard protocols. For qRT-PCR experiments, the cells were differentiated using the mild protocol, which allows for more sensitive assessment of compounds promoting the differentiation process at Day 4 of differentiation (57). 2 d post-confluence, the cells were switched to DMEM supplemented with 10% FBS, 1% P/S, 0.5 mM isobutylmethylxanthine, 0.5 $\mu$M dexamethasone, and 2 $\mu$g/ml insulin for 2 d, with the addition of either 1 $\mu$M Rosi, 8 mM $\beta$OHB, or 8 mM Butyrate. After 2 d, the medium was changed to DMEM supplemented with 10% FBS, 1% P/S, 2 $\mu$g/ml insulin, and the tested compounds for another 2 d, before cells were harvested for RNA isolation. Primary adipocytes from SVF (isolation described below) were cultured like 3T3-L1 cells (described hereafter) and $\beta$OHB and Butyrate were added at 5 mM. For colorimetric analysis of $\beta$OHB utilization by cells, medium was collected on Day 10. $\beta$OHB was determined using the $\beta$-hydroxybutyrate assay kit from Sigma-Aldrich (#MAK041) following the manufacturer's protocol.

C2C12 skeletal muscle cells were cultured in DMEM supplemented with 20% FBS (growth medium, GM) and induced to differentiate with DMEM supplemented with 2% horse serum (HS) (differentiation medium, DM) upon reaching confluence in the presence of either 5 mM $\beta$OHB or 5 mM Butyrate. DM was renewed every other day. Myotube formation was complete (visually) by Day 5.

THP-1 cells were cultured in RPMI 1640 + heat-inactivated FBS and 1% P/S. Differentiation to macrophages was induced with 62.5 ng/ml phorbol 12-myristate 13-acetate (PMA; Sigma-Aldrich) for 24 h in the presence of either 8 mM $\beta$OHB or butyrate.

Human primary monocytes were isolated from buffycoat blood (Sanquin) using the Miltenyi magnet system (CD14 positive selection) and differentiated with 50 ng/ml macrophage colony-stimulating factor (M-CSF) for 7 d. 2 mM butyrate and 2 mM $\beta$OHB were supplied to the differentiation medium from Day 0. Cells were collected at Day 7 for qRT-PCR. Cells were cultured in the RPMI medium and supplemented with 10% FCS, 1% P/S, and 1% GlutaMAX (Gibco, Thermo Fisher Scientific).

Microscopic pictures were taken and cells were subsequently frozen for RNA isolation. All cells were cultured at 37°C with 5% $CO_2$.

### Isolation and differentiation of stromal vascular fraction

Inguinal white adipose tissue from 3 to 4 WT-C57Bl/6 male mice was collected and placed in DMEM (Lonza) supplemented with 1% Penicillin/Streptomycin (PS) and 1% BSA (Sigma-Aldrich). Material was minced finely with scissors and digested in collagenase-containing medium (DMEM with 3.2 mM $CaCl_2$, 1.5 mg/ml collagenase type II (C6885; Sigma-Aldrich), 10% FBS, 0.5% BSA, and 15 mM Hepes) for 1 h at 37°C, with occasional vortexing. Cells were filtered through a 100-$\mu$m cell strainer (Falcon). Subsequently, the cell suspension was centrifuged at 500$g$ for 10 min and the pellet was resuspended in erythrocyte lysis buffer (155 mM $NH_4Cl$, 12 mM $NaHCO_3$, and 0.1 mM EDTA). Upon incubation for 2 min at room temperature, the cells were centrifuged at 500$g$ for 5 min and the

pelleted cells were resuspended in DMEM containing 10% FBS and 1% PS (DMEM/FBS/PS) and plated. Upon confluence, the cells were differentiated according to the protocol as described previously (58, 59). Briefly, confluent SVFs were plated in 1:1 surface ratio, and differentiation was induced 2 d afterwards by switching to a differentiation induction cocktail (DMEM/FBS/PS, 0.5 mM iso-butylmethylxanthine, 1 μM dexamethasone, 7 μg/ml insulin, and 1 μM rosiglitazone) for 3 d. Subsequently, cells were maintained in DMEM/FBS/PS, and 7 μg/ml insulin for 3–6 d and switched to DMEM/FBS/PS for 3 d. Average rate of differentiation was at least 80% as determined by eye.

### Isolation and differentiation of bone marrow derived monocytes

Bone marrow cells were isolated from femurs of WT-C57Bl/6 male mice following the standard protocol and differentiated into macrophages (BMDMs) in 6–8 d in DMEM/FBS/PS supplemented with 20% L929-conditioned medium (L929). After 6–8 d, non-adherent cells were removed, and adherent cells were washed and plated in 12-well plates in DMEM/FBS/PS + 10% L929. After 24 h, medium was switched to 2% L929 in DMEM/FBS/PS overnight. Cells were treated the following day.

### Isolation and differentiation of skeletal myocytes

Myoblasts from hindlimb muscle of WT-C57Bl/6 male mice were isolated as previously described (60). In brief, the muscles were excised, washed in 1× PBS, minced thoroughly, and digested using 1.5 ml collagenase digestion buffer (500 U/ml or 4 mg/ml collagenase type II [C6885; Sigma-Aldrich], 1.5 U/ml or 5 mg/ml Dispase II [D4693; Sigma-Aldrich], and 2.5 mM CaCl2 in 1× PBS) at 37°C water bath for 1 h in a 50 ml tube, agitating the tube every 5 min. After digestion, the cell suspension containing small pieces of muscle tissue was diluted in proliferation medium (PM: Ham's F-10 Nutrient Mix [#31550023; Thermo Fisher Scientific] supplemented with 20% fetal calf serum, 10% HS, 0.5% sterile filtered chicken embryo extract [#092850145; MP Biomedicals], 2.5 ng/ml basic fibroblast growth factor [#PHG0367; Thermo Fisher Scientific], 1% gentamycin, and 1% PS), and the suspension was seeded onto Matrigel-coated (0.9 mg/ml, #354234; Corning) T150 flasks at 20% surface coverage. Cells were grown in 5% CO₂ incubator at 37°C. Confluence was reached latest after 5 d in culture, upon which cells were trypsinized (0.25% trypsin), filtered with 70-μm filters, centrifuged at 300g for 5 min, and then seeded on an uncoated T150 flask for 45 min to get rid of fibroblasts. Subsequently, myoblasts were seeded in PM at 150,000 cells/ml onto Matrigel-coated 12-well plates cells. Upon reaching confluence, differentiation was induced by switching to differentiation medium (DM: Ham's F-10 Nutrient Mix supplemented with 5% HS and 1% PS). DM was replaced every other day. Myotubes fully differentiated by Day 5 of differentiation in DM. The medium was renewed every other day.

### Isolation and culturing of hepatocytes

Primary hepatocytes were isolated from C57BL/6NHsd male mice via collagenase perfusion as described previously (61, 62, 63, 64). Cells were plated onto collagen (0.9 mg/ml) coated 24-well plates at 200,000 cells/well in Williams E medium (PAN Biotech), substituted with 10% FBS, 100 nM dexamethasone, and penicillin/streptomycin and maintained at 37°C in an atmosphere with 5% CO₂. After 4 h of attachment, cells were washed with PBS and allowed to rest in dexamethasone-free medium overnight before treatment.

### Treatments for sequencing experiments

Primary cells were treated for 6 h with 5 mM βOHB or Butyrate, with PBS as control. Adipocytes and Macrophages were treated in DMEM/FCS/PS. Myotubes were treated in DM. Hepatocytes were treated in Williams E medium. Cells were washed with PBS once and stored in –80°C until RNA was isolated.

### RNA isolation & RNA sequencing

Total RNA from all cell culture samples was extracted using TRIzol reagent (Thermo Fisher Scientific) and purified using the QIAGEN RNeasy Mini kit (QIAGEN) according to the manufacturer's instructions. RNA concentration was measured with a NanoDrop 1000 spectrometer and RNA integrity was determined using an Agilent 2100 Bioanalyzer with RNA 6000 microchips (Agilent Technologies). Library construction and RNA sequencing on BGISEQ-500 were conducted at Beijing Genomics Institute (BGI) for pair-end 150 bp runs. At BGI, genomic DNA was removed with two digestions using amplification grade DNAse I (Invitrogen). The RNA was sheared and reverse transcribed using random primers to obtain cDNA, which was used for library construction. The library quality was determined by using Bioanalyzer 2100 (Agilent Technologies). Then, the library was used for sequencing with the sequencing platform BGISEQ-500 (BGI). All the generated raw sequencing reads were filtered, by removing reads with adaptors, reads with more than 10% of unknown bases, and low quality reads. Clean reads were then obtained and stored as FASTQ format.

The RNA-seq reads were used to quantify transcript abundances. To this end the tool *Salmon* (65) (version 1.2.1) was used to map the reads to the GRCm38.p6 mouse genome assembly-based transcriptome sequences as annotated by the GENCODE consortium (release M25) (66). The obtained transcript abundance estimates and lengths were then imported in R using the package *tximport* (version 1.16.1) (67), scaled by average transcript length and library size, and summarized on the gene-level. Such scaling corrects for bias due to correlation across samples and transcript length and has been reported to improve the accuracy of differential gene expression analysis (67). Differential gene expression was determined using the package *limma* (version 3.44.3) (68) using the obtained scaled gene-level counts. Briefly, before statistical analyses, nonspecific filtering of the count table was performed to increase detection power (69), based on the requirement that a gene should have an expression level greater than 20 counts, that is, one count per million reads (cpm) mapped, for at least six libraries across all 36 samples. Differences in library size were adjusted by the trimmed mean of M-values normalization method (70). Counts were then transformed to log-cpm values and associated precision weights, and entered into the *limma* analysis pipeline (71). Differentially expressed genes were identified by using

generalized linear models that incorporate empirical Bayes methods to shrink the standard errors towards a common value, thereby improving testing power ([68](), [72]()). Genes were defined as significantly changed when $P < 0.001$.

### Biological interpretation of transcriptome data RNA isolation & RNA sequencing

Changes in gene expression were related to biologically meaningful changes using GSEA ([73]()). It is well accepted that GSEA has multiple advantages over analyses performed on the level of individual genes ([73](), [74](), [75]()). GSEA evaluates gene expression on the level of gene sets that are based on prior biological knowledge, for example, published information about biochemical pathways or signal transduction routes, allowing more reproducible and interpretable analysis of gene expression data. As no gene selection step (fold-change and/or $P$-value cutoff) is used, GSEA is an unbiased approach. A GSEA score is computed based on all genes in gene set, which boosts the signal-to-noise ratio and allows to detect affected biological processes that are due to only subtle changes in expression of individual genes. This GSEA score called normalized enrichment score can be considered as a proxy for gene set activity. Gene sets were retrieved from the expert-curated KEGG pathway database ([76]()). Only gene sets comprising more than 15 and fewer than 500 genes were taken into account. For each comparison, genes were ranked on their t-value that was calculated by the moderated $t$ test. Statistical significance of GSEA results was determined using 10,000 permutations.

### Quantitative real-time PCR

Reverse transcription was performed using the iScript cDNA Synthesis Kit (Bio-Rad) according to the manufacturer's protocol using 250 ng RNA for in vitro studies. Quantitative PCR amplifications were carried out on a CFX 384 Bio-Rad thermal cycler (Bio-Rad) using SensiMix PCR reagents (Bioline, GC Biotech). Gene expression values were normalized to one of the housekeeping genes and analyzed using delta $\Delta\Delta$Ct method. Primer sequences of genes are provided in Table S4.

### Animal approval

Animals for primary cell experiments were all housed at the Centre for Small Animals, which is part of the Centralized Facilities for Animal Research at Wageningen University and Research (CARUS) and were approved by the Local Animal Ethics Committee of Wageningen University (AVD104002015236: 2016.W-0093.005, 2016.W-0093.007). Mice were maintained at 21°C, on rodent chow and kept on a regular day–night cycle (lights on from 6:00 AM to 6:00 PM).

Some primary cells were obtained in Munich. All animal studies were conducted in accordance with German animal welfare legislation. Male C57BL/6N mice obtained from Charles River laboratories were maintained in a climate-controlled environment with specific pathogen-free conditions with 12-h dark/light cycles in the animal facility of the Helmholtz Centre. Protocols were approved by the Institutional Animal Welfare Officer, and necessary licenses were obtained from the state ethics committee and government of Upper Bavaria (55.2-1-54-2532.0-40-15). Mice were fed ad libitum with regular rodent chow.

### Statistical analyses

Statistical analysis of the transcriptomics data was performed as described in the previous paragraph. Data are presented as mean ± SD. $P$-values < 0.05 were considered statistically significant.

## Data Availability

The RNAseq data from publication have been deposited to the GEO database and assigned the accession number: GSE179023.

## Supplementary Information

## Acknowledgements

We thank all the members of the Nutrition, Metabolism and Genomics group for fruitful discussions. We gratefully acknowledge the assistance of Shohreh Keshtkar for the qRT-PCR measurements. This work was supported by the Netherlands Heart Foundation (CVON ENERGISE grant CVON2014-02).

### Author Contributions

PMM Ruppert: conceptualization, formal analysis, validation, investigation, visualization, methodology, project administration, and writing—original draft, review, and editing.
L Deng: conceptualization, resources, formal analysis, investigation, visualization, methodology, and writing—review and editing.
GJEJ Hooiveld: resources, data curation, formal analysis, visualization, and writing—review and editing.
RWJ Hangelbroek: resources, data curation, formal analysis, visualization, and writing—review and editing.
A Zeigerer: resources, methodology, and writing—review and editing.
S Kersten: conceptualization, resources, formal analysis, supervision, funding acquisition, validation, project administration, and writing—review and editing.

### Conflict of Interest Statement

The authors declare that they have no conflict of interest.

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
