## [Reviewer comments · Life Science Alliance]

Life Science Alliance

RNA-sequencing reveals niche gene expression effects of beta-hydroxybutyrate in primary myotubes

Philip Ruppert, Lei Deng, Guido Hooiveld, Roland Hangelbroek, Anja Zeigerer, and Sander Kersten
DOI: <https://doi.org/10.26508/lsa.202101037>

Corresponding author(s): Sander Kersten, Wageningen University

Review Timeline:	Submission Date:	2021-01-26
	Editorial Decision:	2021-03-26
	Revision Received:	2021-06-29
	Editorial Decision:	2021-07-28
	Revision Received:	2021-07-29
	Accepted:	2021-07-30

Transaction Report:

March 26, 2021

Re: Life Science Alliance manuscript #LSA-2021-01037

Prof. Sander Kersten
Wageningen University
Human Nutrition
Bomenweg 2
Wageningen 6703HD
Netherlands

Dear Dr. Kersten,

Thank you for submitting your manuscript entitled "RNA-sequencing reveals niche gene expression effects of beta-hydroxybutyrate in primary myotubes" to Life Science Alliance. The manuscript was assessed by expert reviewers, whose comments are appended to this letter.

We apologize for this extended and unusual delay in getting back to you. As you will note from the reviewers' comments below, both reviewers are quite interested and intrigued by these findings. While Reviewer 2 does not have any additional requests, Reviewer 1 has made some reasonable requests for validation experiments, which, in our opinion, should be included in the revised manuscript. We would, thus, like to invite you to submit a revised version of this manuscript that addresses all of Reviewer 1's points.

Thank you for this interesting contribution to Life Science Alliance. We are looking forward to receiving your revised manuscript.

Sincerely,

Shachi Bhatt, Ph.D.

Executive Editor

Life Science Alliance

<https://www.lsa-journal.org/>

Interested in an editorial career? EMBO Solutions is hiring a Scientific Editor to join the international Life Science Alliance team. Find out more here -

https://www.embo.org/documents/jobs/Vacancy_Notice_Scientific_editor_LSA.pdf

B. MANUSCRIPT ORGANIZATION AND FORMATTING:

Reviewer #1 (Comments to the Authors (Required)):

Dear Editor, Dear Authors:

Rupper and collaborators present a well performed study in which

- 1) the modulatory properties of Butyrate (But) and the ketone body beta-hydroxybutyrate (BHB) on transcription are investigated by RNAseq on mouse derived (and differentiated) primary cells
- 2) The effects of But and BHB on differentiation of adipocytes, muscles and monocytes are investigated using cell lines.

The overall findings of the manuscript support the idea that i) BHB does not impact the cell differentiation of the lineages mentioned above and ii) the impact of BHB on the transcriptome is rather limited (if not negligible) on the cell type studied, except for the muscle lineage.

The manuscript is interesting inasmuch it adds new concepts, mainly gathered from transcriptional analysis by RNAseq, on the rather controversial field investigating the action of BHB.

I wish to present to the authors a number of commentaries:

- In the first part of the manuscript, the effect of But - but not BHB - on inhibiting differentiation in the cell lines (THP, 3T3L1, C2C12) is shown. Here, But and BHB treatments are "chronic", i.e. done throughout differentiation. On the opposite, the RNA-seq on differentiated primary cells from mice is done after a 6-h But / BHB treatment. It would be interesting, for the reader, to know whether the effect of But (and lack of effect of BHB) on differentiation would also occur on the primary cultures from mouse-derived tissues.
- But treatment of cells readily induces histone acetylation and BHB is known to promote beta-hydroxybutyrylation (with acetylation remaining controversial). A quite necessary experiment would be western blotting on histones to show But-induced acetylation and BHB-induced beta-hydroxybutyrylation in the samples submitted to RNA-seq. I appreciate that the authors show the existence of monocarboxylate transporters (MCTs) on their RNA-seq readings, but at the same time the occurrence of acetylation and (most important) beta-hydroxybutyrylation would give a biochemical proof that But and BHB enter the cells, especially in the context of the very little transcriptional effects of BHB in all cells except myotubes.
- The authors state that the few genes modulated by BHB in "BHB-unresponsive" cells might be false positives. In my opinion, this over-reliance on bioinformatics and statistics may be misleading. As a general rule, a classical qPCR quantification of the few most overexpressed / repressed genes would be welcome, both for BHB-unresponsive cells as well as for muscle cells.
- Legend to figure 1 is not complete as also THP-1/macrophages are investigated
- Figure 2A, reporting RNA-seq reads of Mct genes, GAPDH and BDH1 would benefit of statistical analysis and, importantly, also of qPCR validation/replication
- In figure 2D, the replicates should be identified.
- Supplemental figure 2 (panels A-C) and its legend (reporting panels A to D) are discordant. Please amend.
- Please be sure to identify in the manuscript the platform/database in which RNA-seq data will be deposited (GEO or other), as I believe that that availability of raw data will be of interest for the scientific community working on BHB.
- When re-submitting the manuscript, please make sure to number the pages and please compress the files (the full ms PDF of the first submission is 173164KB !)

Reviewer #2 (Comments to the Authors (Required)):

The manuscript submitted by Sander Kersten and co-workers presents the data on the beta-hydroxybutyrate (3HB), the main transport form of ketone bodies, and its potential as a signaling molecule.

Using different primary cell types: adipocytes, macrophages, myotubes the authors studied the effect of 3HB on in vitro differentiation process, but also capacity of the tested ketone body for modification of gene expression profile of primary mouse adipocytes, macrophages, myotubes and

hepatocytes via genome wide sequencing - RNAseq.

It was found that 3HB does not affect polarization of myotubes as well as differentiation of monocytes THP-1 and adipocytes. Also RNAseq analysis revealed minimal impact of 3HB on gene expression profile of the analyzed cells. The biggest modifications were found in the myocytes.

The presented study gives a valuable input into the of 3HB biochemistry, supplementing the current state-of-the-art ketone bodies focused in the substantial amount of transcriptomic data. The paper is of very good quality. The introduction is coherent and very well brings to the topic of the paper, describing do far identified effects exerted by 3HB, including controversies. The methods for validation the hypothesis are properly chosen and described in sufficient amount of details. The studies performed, are properly designed and controlled, replicated suitable. Statistical analysis of data is performed properly. The conclusions reached, are consistent with the presented data.

Point-by point response for revised manuscript LSA-2021-01037

Editor and Reviewer comments:

Reviewer 1:

Ruppert and collaborators present a well performed study in which

- 1) the modulatory properties of Butyrate (But) and the ketone body beta-hydroxybutyrate (BHB) on transcription are investigated by RNAseq on mouse derived (and differentiated) primary cells
- 2) The effects of But and BHB on differentiation of adipocytes, muscles and monocytes are investigated using cell lines.

The overall findings of the manuscript support the idea that i) BHB does not impact the cell differentiation of the lineages mentioned above and ii) the impact of BHB on the transcriptome is rather limited (if not negligible) on the cell type studied, except for the muscle lineage.

The manuscript is interesting inasmuch it adds new concepts, mainly gathered from transcriptional analysis by RNAseq, on the rather controversial field investigating the action of BHB.

I wish to present to the authors a number of commentaries:

- In the first part of the manuscript, the effect of But - but not BHB - on inhibiting differentiation in the cell lines (THP, 3T3L1, C2C12) is shown. Here, But and BHB treatments are "chronic", i.e. done throughout differentiation. On the opposite, the RNA-seq on differentiated primary cells from mice is done after a 6-h But / BHB treatment. It would be interesting, for the reader, to know whether the effect of But (and lack of effect of BHB) on differentiation would also occur on the primary cultures from mouse-derived tissues.

Our response: We thank the reviewer for this suggestion. We replicated the differentiation experiments using primary adipocytes and primary human monocytes. For the latter we had to reduce the concentration of bOHB/Butyrate to 2 mM, as it killed the cells very rapidly at 5 or 8 mM. It was observed that the inhibition of differentiation by butyrate was confirmed in both primary cell types (Figure 1G-I). qPCR reflected reductions in expression of differentiation markers (Figure 1G, I). As observed in immortalized cells, bOHB did not impact differentiation in primary cells (Figure 1G, I).

- But treatment of cells readily induces histone acetylation and BHB is known to promote beta-hydroxybutyrylation (with acetylation remaining controversial). A quite necessary experiment would be western blotting on histones to show But-induced acetylation and BHB-induced beta-hydroxybutyrylation in the samples submitted to RNA-seq. I appreciate that the authors show the existence of monocarboxylate transporters (MCTs) on their RNA-seq readings, but at the same time the occurrence of acetylation and (most important) beta-hydroxybutyrylation would give a

biochemical proof that But and BHB enter the cells, especially in the context of the very little transcriptional effects of BHB in all cells except myotubes.

Our response: We thank the reviewer for this valuable remark. We indeed did not present formal evidence that BHB has entered the cells in our experiments. Due to the complicated nature of setting up novel antibodies and isolation of histones we opted to assess utilization of bOHB by measuring the decline in bOHB concentration in the cell culture medium over time. Our findings suggest that primary adipocytes utilized about 50% of the bOHB in the medium after 3 days of culturing. These data are shown in Supplemental figure 2B.

Taking into account the consistent inhibitory effects of butyrate in our differentiation experiments as well as the massive gene expression effects in our RNA-sequencing experiment, we think that it is highly improbable that butyrate does not interact with cells.

- The authors state that the few genes modulated by BHB in "BHB-unresponsive" cells might be false positives. In my opinion, this over-reliance on bioinformatics and statistics may be misleading. As a general rule, a classical qPCR quantification of the few most overexpressed / repressed genes would be welcome, both for BHB-unresponsive cells as well as for muscle cells.

We thank the reviewer for this request. We have performed qPCR analysis on exclusively upregulated by butyrate (Dgat2, H1f0), exclusively downregulated by butyrate (Inpp1 and Phlda3) as well as genes exclusively downregulated by bOHB in myocytes (Clec4a1 and Apoc2). The qPCR results confirm the RNA-seq data and are shown in Supplemental figure 2F,G.

- Legend to figure 1 is not complete as also THP-1/macrophages are investigated

We thank the reviewer for this request. We have amended the legend of figure 1.

- Figure 2A, reporting RNA-seq reads of Mct genes, GAPDH and BDH1 would benefit of statistical analysis and, importantly, also of qPCR validation/replication

Done. Validation of the expression of ketolysis-related genes are shown in Supplemental figure 2A.

- In figure 2D, the replicates should be identified.

Done.

- Supplemental figure 2 (panels A-C) and its legend (reporting panels A to D) are discordant. Please amend.

We thank the reviewer for pointing this out and amended it accordingly.

-Please be sure to identify in the manuscript the platform/database in which RNA-seq data will be

deposited (GEO or other), as I believe that that availability of raw data will be of interest for the scientific community working on BHB.

We thank the reviewer for this request. We've deposited the dataset under GEO accession number GSE179023.

- When re-submitting the manuscript, please make sure to number the pages and please compress the files (the full ms PDF of the first submission is 173164KB !)

We thank the reviewer for pointing this out. We adjusted both issues in our resubmission.

Reviewer #2 (Comments to the Authors (Required)):

The manuscript submitted by Sander Kersten and co-workers presents the data on the beta-hydroxybutyrate (3HB), the main transport form of ketone bodies, and its potential as a signaling molecule.

Using different primary cell types: adipocytes, macrophages, myotubes the authors studied the effect of 3HB on in vitro differentiation process, but also capacity of the tested ketone body for modification of gene expression profile of primary mouse adipocytes, macrophages, myotubes and hepatocytes via genome wide sequencing - RNAseq.

It was found that 3HB does not affect polarization of myotubes as well as differentiation of monocytes THP-1 and adipocytes. Also RNAseq analysis revealed minimal impact of 3HB on gene expression profile of the analyzed cells. The biggest modifications were found in the myocytes.

The presented study gives a valuable input into the of 3HB biochemistry, supplementing the current state-of-the-art ketone bodies focused in the substantial amount of transcriptomic data. The paper is of very good quality. The introduction is coherent and very well brings to the topic of the paper, describing do far identified effects exerted by 3HB, including controversies. The methods for validation the hypothesis are properly chosen and described in sufficient amount of details. The studies performed, are properly designed and controlled, replicated suitable. Statistical analysis of data is performed properly. The conclusions reached, are consistent with the presented data.

We thank the reviewer for reviewing our manuscript and the positive comments.

July 28, 2021

RE: Life Science Alliance Manuscript #LSA-2021-01037R

Prof. Sander Kersten
Wageningen University
Human Nutrition
Bomenweg 2
Wageningen 6703HD
Netherlands

Dear Dr. Kersten,

Thank you for submitting your revised manuscript entitled "RNA-sequencing reveals niche gene expression effects of beta-hydroxybutyrate in primary myotubes". We would be happy to publish your paper in Life Science Alliance pending final revisions necessary to meet our formatting guidelines.

- please consult our manuscript preparation guidelines <https://www.life-science-alliance.org/manuscript-prep> and make sure your manuscript sections are in the correct order
- please add your main, supplementary figure, and table legends to the main manuscript text after the references section
- please upload your Tables in editable .doc or excel format
- there is a callout in the manuscript text for Figure S1A although the actual figure and its legend doesn't have panels. Please correct
- please add an Approval Statement indicating approval for the work using mice

Figure checks:

- missing scale bars for figure 1A, C, E, and H, please indicate their size in the Figure Legend

LSA now encourages authors to provide a 30-60 second video where the study is briefly explained. We will use these videos on social media to promote the published paper and the presenting author. Corresponding or first-authors are welcome to submit the video. Please submit only one video per manuscript. The video can be emailed to contact@life-science-alliance.org

A. FINAL FILES:

B. MANUSCRIPT ORGANIZATION AND FORMATTING:

Sincerely,

Reviewer #1 (Comments to the Authors (Required)):

Dear Monitoring Editor, Dr. Cooper,
Dear authors

firstly, please accept my apologies and pass them to the authors for the delay in the submission of my review.

I read with interest the rebuttal and revised version of the manuscript by Ruppert and collaborators;

Overall, it is my opinion that the authors did a thoughtful and fully satisfactory revision work that addressed almost all the points that I raised. I hope the authors will share my opinion that the manuscript has been improved, and will have a good visibility and importance in the field.

I noticed that the authors fell short of providing evidence for the occurrence of histone-beta hydroxybutyrylation. I really would have liked to see these data included. The measurement of BHB in the medium, serving as a reflection that the molecule is metabolized by the cells, does not formally prove that histone beta hydroxybutyrylation takes place. The authors claim that antibody production is cumbersome and time consuming, which is true. Alternatively, the authors may have considered using the commercial antibodies marketed by PTM biolabs.

In spite of the absence of these blots, the manuscript is more than worth accepting, and I have no doubts that histone beta hydroxybutyrylation is taking place (it is published elsewhere at least for some of the cell types used in this investigation), but providing these blots would have been the cherry on the cake.

I congratulate the authors for this very interesting work, and present again my apologies for my delay in responding to the resubmitted version.

Sincerely, Reviewer 1

July 30, 2021

RE: Life Science Alliance Manuscript #LSA-2021-01037RR

Prof. Sander Kersten
Wageningen University
Human Nutrition
Bomenweg 2
Wageningen 6703HD
Netherlands

Dear Dr. Kersten,

Thank you for submitting your Research Article entitled "RNA-sequencing reveals niche gene expression effects of beta-hydroxybutyrate in primary myotubes". It is a pleasure to let you know that your manuscript is now accepted for publication in Life Science Alliance. Congratulations on this interesting work.

DISTRIBUTION OF MATERIALS:

Again, congratulations on a very nice paper. I hope you found the review process to be constructive and are pleased with how the manuscript was handled editorially. We look forward to future exciting submissions from your lab.

Sincerely,
